# Exposure of gut bacterial isolates to the anthelminthic drugs, ivermectin and moxidectin, leads to antibiotic-like phenotypes of growth inhibition and adaptation

Julian Dommann [1,2], Jennifer Keiser [1,2], Julian Garneau [3], Alison Gandelin[3], Carlo Casanova[4], Peter M. Keller[5], Somphou Sayasone[1,2,6], Pascale Vonaesch [3] & Pierre H. H. Schneeberger [1,2] ✉

Due to their broad-spectrum activities, ivermectin and moxidectin are widely used anthelminthics in veterinary and human medicine. However, ivermectin has recently been shown to perturbate bacterial growth. Given the macrolide-like structure of both ivermectin and moxidectin, there is a need to characterize the antibiotic spectrum of these anthelminthics and their potential implications in the development of cross-resistance to macrolides and other families of antibiotics. Here, we characterize growth dynamics of 59 bacterial isolates in presence of ivermectin and moxidectin. Further, we assessed the effect of repeated anthelminthic exposure in 5 bacterial isolates on sensitivity to different antibiotics, both via growth dynamics and minimal inhibitory concentration. We found, that anthelminthic growth phenotypes are comparable to a selection of tested antibiotics. Bacterial anthelminthic challenging resulted in decreased anthelminthic sensitivity, and to some extent, decreased antibiotic sensitivity. Hence, potential off-target effects of large-scale administration of ivermectin and moxidectin should be carefully monitored.

## Background

Soil-transmitted helminths (STHs) are a multi-species group of parasitic, eukaryotic worms residing in species-specific niches within the host's gastrointestinal (GI) tract. Amongst all neglected tropical diseases listed by the World Health Organization (WHO), STH infections dominate in terms of prevalence (over 1.5 billion) and disease burden[1,2], mainly affecting preschool and school-aged children and potentially leading to serious long-term health conditions, such as impairment in cognitive and physical development. While the improvement of water quality, sanitation and hygiene represents a long-term countermeasure, the short-term remedy relies on anthelminthic drugs to reduce morbidity[3–5]. The current frontline treatment encompasses the two benzimidazoles mebendazole and albendazole (ALB)[6]. However, cure rates vary in a species-specific manner[7,8].

Alarmingly, single-dose regimens of either drug yield poor cure rates in *Trichuris trichiura* infections and single-dose regimens of mebendazole result in low cure rates against hookworm (*Ancylostoma duodenale, Necator americanus*) infections[7,9,10]. As the pipeline for new drugs remains scarcely populated, the focus currently lies on repurposed veterinary drugs[11] and combination therapies[7,8].

Ivermectin (IV) and moxidectin (MX) are two prime examples of such repurposed drugs. A 16-membered macrocyclic lactone ring, the chemical structure that represents the group of macrolide antibiotics, characterizes both IV and MX. Macrolide antibiotics such as erythromycin (EM), clarithromycin (CH) or azithromycin (AZ) interfere with bacterial protein synthesis[12] and are typically used in treatment of gram-positive bacterial infections. In contrast, IV and MX possess a broad activity spectrum against

[1]Department of Medical Parasitology and Infection Biology, Swiss Tropical and Public Health Institute, Allschwil, Switzerland. [2]University of Basel, Basel, Switzerland. [3]Department of Fundamental Microbiology, University of Lausanne, Lausanne, Switzerland. [4]Institute for Infectious Diseases, University of Bern, Bern, Switzerland. [5]Clinical Bacteriology and Mycology, University Hospital Basel, Basel, Switzerland. [6]Lao Tropical and Public Health Institute, Ministry of Health, Vientiane, Lao People's Democratic Republic. ✉e-mail: pierre.schneeberger@swisstph.ch

numerous parasites and are thus frequently administered as antiparasitic agents. MX was recently approved to treat human onchocerciasis[13] and was shown to have a good efficacy against *Strongyloides stercoralis*[14,15] - and in combination with ALB - against *T. trichiura*[16]. IV on the other hand, is a widely used anthelminthic for filarial infections and used in combination therapy paired with ALB to increase treatment efficacy against *T. trichiura* infections[17], an indication recently added the WHO's Model List of Essential Medicine[18].

As orally administered drugs pass through the small and large intestine, they are subject to interactions with numerous microbes before being absorbed, making the gut microbiome a relevant site for first-pass metabolism[19]. The pan-genome of gut microbes encompasses 150-fold more genes compared to their host, resulting in outstanding genetic diversity and therefore considerable possibilities of direct and indirect interactions that may influence bioavailability and efficacy of an administered compound[20].

Vice versa, drugs or their metabolites could also promote or inhibit bacterial growth, since antibacterial properties are not limited to antibiotic drugs[21]. Hence, orally administered non-antibiotic drugs could perturb gut-microbial growth[21] and thus disrupt gut microbiome homeostasis. Due to the molecular structure of IV and MX, we hypothesize that various gut bacterial isolates could be sensitive to these drugs. Indeed, a recent study found an association between pre-treatment gut microbial composition and treatment outcome in participants receiving an IV-ALB combination therapy against *T. trichiura* infections[22]. Treatment failure was associated with gut bacterial species, such as *Prevotella copri*, *Streptococcus salivarius* and *Faecalibacterium prausnitzii*—among others[22]. Moreover, repeated IV or MX exposure could influence bacterial sensitivity profiles, which potentially leads to cross-resistant gut bacterial isolates or interferes with IV/MX treatment efficacy – depending on the underlying adaptation mechanism.

To our knowledge, only limited data regarding antibacterial activity of IV and MX against gut bacterial isolates have been published to date[21,23]. A causal relationship between bacterial sensitivity profiles and IV or MX treatment would raise serious concerns in light of the extensive use of IV and MX in the livestock industry and the increase in distribution and administration for helminth infections and malaria transmission control[24–26]. Thus, it is crucial to anticipate potential off-target effects of these two anthelminthics. In light of previously published studies, we, therefore, aimed to characterize antibacterial properties of these two widely used anthelminthics in-depth against a broad range of gut bacterial isolates in vitro, especially in the context of cross-reaction with other antibiotics.

## Results
### IV/MX inhibit in vitro growth of a broad range of gut bacterial isolates

To explore whether and to what extent IV/MX inhibit growth of gut bacteria, we conducted a series of in vitro experiments, incubating a broad taxonomic range of gut bacterial isolates in different concentrations of IV/MX. Specifically, we worked with 59 aerobically and anaerobically grown bacterial species (Supplementary Tables 1–3) across 10 genera (*Actinomyces, Bacteroides, Blautia, Clostridium, Dorea, Enterococcus, Escherichia, Lactobacillus, Staphylococcus, Streptococcus*). A detailed rationale covering the selection of isolates can be found in the Methods section. After oral administration, IV/MX immediately reach the target tissue for helminth infections– the gut linen–, resulting in estimated physiological concentration levels of ~5 μM in the intestine[21]. We therefore incubated isolates with IV and MX at 1 μM, 5 μM and 10 μM (Supplementary Figs. 2–7). To compare growth curves across isolates and individual assays, we adapted the concept of area under the curve (AUC) ratios, as published by Gencay et al.[27]. Namely, we divided the AUC in presence of IV or MX at a given concentration, by the AUC of the corresponding control curve.

Overall, a drug concentration of 1 μM IV/MX does not appear to have a negative impact on growth of any of the bacteria tested (Fig. 1A, B). Inter-species comparison amongst aerobically incubated isolates reveals that

isolates of the genus *Enterococcus* do not display delayed growth in presence of IV and MX across all tested concentrations (Fig. 1A). In contrast, isolates belonging to the genera *Actinomyces* and *Streptococcus* are sensitive to the presence of IV and MX at 5 and 10 μM. Anaerobically, the tested isolates amongst the genera *Bacteroides, Lactobacillus* and *Staphylococcus* are not affected by the presence of either IV or MX (Fig. 1B). In the contrary, we observed decreased AUC ratios amongst the genera *Blautia, Clostridium, Dorea* and *Streptococcus*. *S. salivarius (01)* and *Streptococcus parasanguinis (01)* were incubated both aerobically and anaerobically and were sensitive to IV/MX in both cases. Intra-species comparison amongst aerobically incubated isolates reveals drug-dependent sensitivity. For instance, *S. pneumoniae (02)* displays lower sensitivity to IV or MX ($AUC_{IV10\mu M}$: 0.84; $AUC_{MX10\mu M}$: 0.47), compared *S. pneumoniae (01)* and *S. pneumoniae (03)* ($AUC_{IV10\mu M}$: 0.47; $AUC_{MX10\mu M}$: 0.1; $AUC_{IV10\mu M}$: 0.1; $AUC_{MX10\mu M}$: 0.1). Anaerobically, we also observed differences in intra-species sensitivity. In fact, 1/3 *Dorea* isolates (*D. longicatena (01)*; $AUC_{IV5\mu M}$: < 0.1) and 1/5 *Clostridium* isolates (*C. baratii (02)*; $AUC_{MX10\mu M}$: < 0.1) were inhibited by either IV or MX.

To transition from quantitative values (AUC ratio) to qualitative values in Fig. 1C, D ("sensitive"; "not sensitive") we employed a tentative AUC ratio cut-off of 0.8 to stratify isolates into "sensitive" and "not sensitive". Pairwise comparison of isolates across the three concentrations resulted in an inverse relationship between drug dose and AUC ratio (dose dependence; Tables 1 and 2).

According to the tentative cut-off, 25/59 isolates were considered sensitive towards MX and 19/59 towards IV. Although there is a general trend of lower AUC ratios with higher drug concentrations for both IV and MX, AUC ratios vary substantially across genera and species in presence of 5 μM or 10 μM IV or MX. Lastly, we also observed a phenotype consisting of a drug-specific response in several isolates. For instance, incubation of *S. anginosus (01)* and *S. dysgalactiae (01)* (Fig. 1A, E) with MX results in lower AUC ratios, compared with IV at the same concentration. In contrast *A. odontolyticus (01), A. odontolyticus (02)* and *D. longicatena (01)* seem to have a higher sensitivity towards IV. Within our set of tested species, more isolates appear to be sensitive towards MX (Fig. 1E, F) and at equal concentrations of IV or MX, MX results in lower AUC ratios (Fig. 1A, B).

### In vitro potency of IV/MX is comparable to a selection of antibiotics

We next sought to compare the potency of the two anthelmintic drugs to that of antibiotic compounds in vitro. Since IV and MX belong to the macrocyclic lactone drug class, we included the three macrolide antibiotics EM, CH, and AZ in the comparison. Clindamycin (CL) represents a lincosamide antibiotic, which shares the mode of action (MOA) with macrolide antibiotics. We further included the fluoroquinolone ciprofloxacin (CX), tetracycline (TC) and the two carbapenems, meropenem (MP) and imipenem (IP), as they possess completely different modes of action. Therefore, we incubated a subset of anthelmintic-susceptible bacterial isolates, comprised of 8 *Streptococcus spp.* and 3 *Actinomyces spp.* identified in our primary screen, with a selection of antibiotics (EM, CH, AZ, CL, CX, TC, MP, IP) at concentrations of 1 μM, 5 μM and 10 μM to simulate the magnitude as in incubations with IV/MX (Supplementary Figs. 8–11). In contrast to IV and MX, the tested antibiotics cause a reduction in AUC ratios at 1 μM (Supplementary Fig. 12, KW; $p = 1.186e$-08, chi-squared = 32.509, df = 1). However, incubations with IV/MX at 5 μM or 10 μM result in AUC ratios in a comparable range as the tested antibiotics (KW; $p = 0.03708$, chi-squared = 4.3467, df = 1 and KW; $p = 0.8284$, chi-squared = 0.046964, df = 1). *S. dysgalactiae (01)* represents an exception at 5 μM, as it displays high AUC ratios for both IV and MX (Supplementary Fig. 13, $AUC_{IV5\mu M}$: 1.06; $AUC_{MX5\mu M}$: 0.95). Similarly, solely *S. dysgalactiae (01)* displays a high AUC ratio for IV at a concentration of 10 μM (Fig. 2, $AUC_{IV10\mu M}$: 0.96). In the case of *S. oralis (02)* and *S. pneumoniae (03)*, IV and MX at 10μM achieved the lowest AUC ratios across all tested drugs ($AUC_{IV10\mu M}$: 0.07; $AUC_{MX10\mu M}$: 0.07 and $AUC_{IV10\mu M}$: 0.06; $AUC_{MX10\mu M}$: 0.06). Results of a pairwise Wilcoxon rank sum test between all compounds can be found in

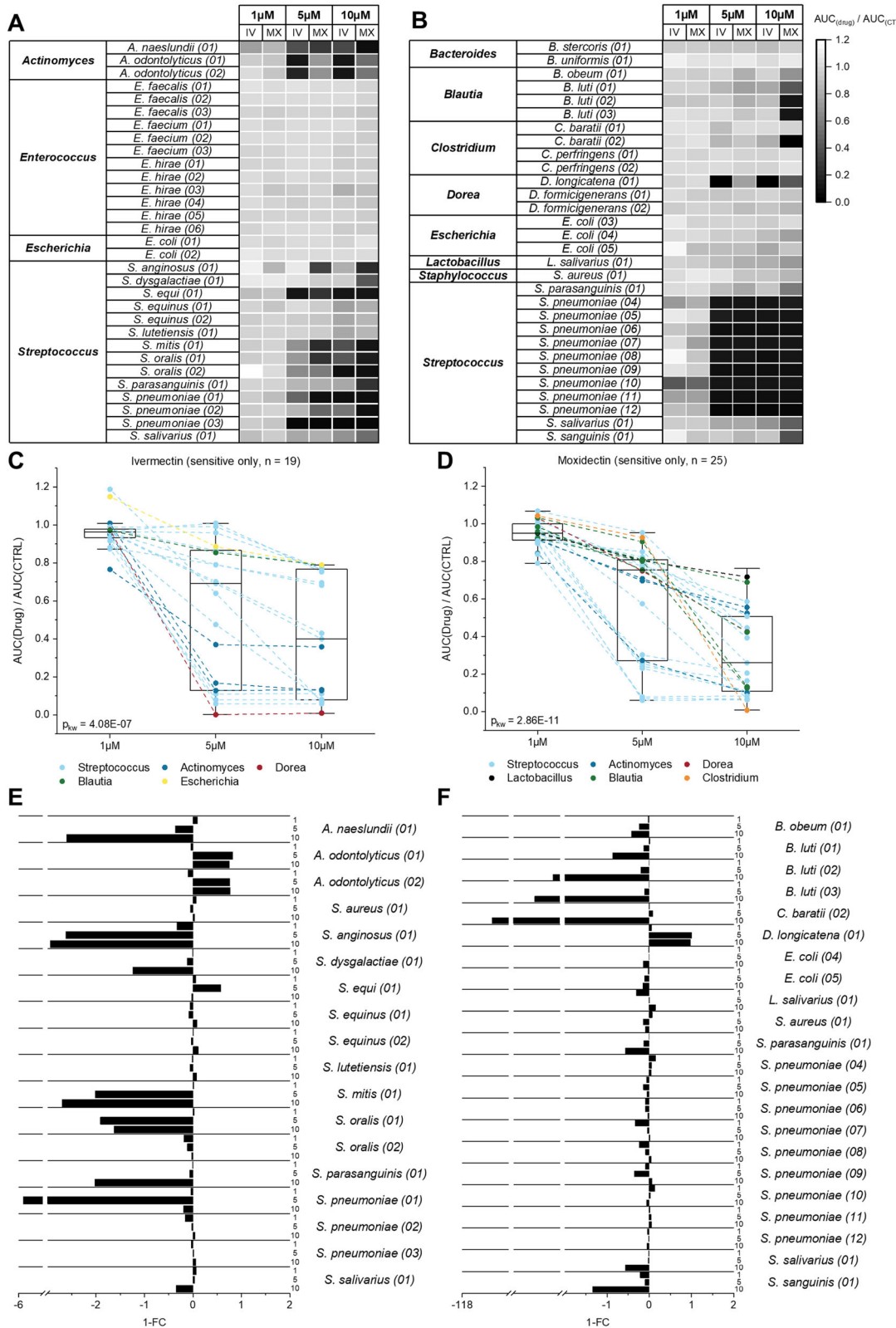

**Fig. 1 | Sensitivity of gut bacterial isolates against ivermectin (IV) and moxidectin (MX).** **A** AUC ratios of aerobically incubated isolates for IV and MX at concentrations of 1 µM, 5 µM and 10 µM. **B** AUC ratios of anaerobically incubated isolates for IV and MX at concentrations of 1 µM, 5 µM and 10 µM. **C** AUC ratios of 25 MX-sensitive isolates visualized across three concentrations (1 µM, 5 µM, 10 µM). Values belonging to the same isolate are connected with a dotted line. Coloring was chosen according to the isolate genus. In the boxplot, the box represents the interquartile range, contains a median line and whiskers indicate the 95%

CI. **D** AUC ratios of 19 IV-sensitive isolates visualized across three concentrations (1 µM, 5 µM, 10 µM). Values belonging to the same isolate are connected with a dotted line. Coloring was chosen according to the isolate genus. In the boxplot, the box represents the interquartile range, contains a median line and whiskers indicate the 95% CI. **E, F** Fold change (FC) ratio for sensitive isolates. FC was calculated as 1−(AUC$_{IV}$ / AUC$_{MX}$). 1−FC > 0 indicates a higher sensitivity towards IV, 1−FC < 0 indicates higher sensitivity towards MX. The figure is stratified by aerobic (**E**) and anaerobic (**F**) incubations.

**Table 1 | Dose dependence in IV incubations**

| | IV | | | |
|---|---|---|---|---|
| Concentrations | 1 µM vs. 5 µM (W) | 5 µM vs.10 µM (W) | 1 µM vs.10 µM (W) | 1 µM vs.5 µM vs.10 µM (KW) |
| p-value | 5.40E-05 | 1.30E-01 | 3.20E-09 | 4.08E-07 (chi-squared = 29.425) |

P values of pairwise Wicoxon rank sum tests (W) and Kruskal–Wallis (KW) test performed on AUC ratios of IV-sensitive bacterial isolates (n = 19) for the incubation concentrations 1 µM, 5 µM and 10 µM (IV = ivermectin).

**Table 2 | Dose dependence in MX incubations**

| | MX | | | |
|---|---|---|---|---|
| Concentrations | 1 µM vs. 5 µM (W) | 5 µM vs.10 µM (W) | 1 µM vs.10 µM (W) | 1 µM vs.5 µM vs.10 µM (KW) |
| p-value | 2.10E-08 | 6.00E-04 | 4.70E-14 | 2.86E-11(chi-squared = 48.557) |

P-values of pairwise Wicoxon rank sum tests (W) and Kruskal–Wallis (KW) test performed on AUC ratios of MX-sensitive bacterial isolates (n = 25) for the incubation concentrations 1 µM, 5 µM and 10 µM (MX = moxidectin).

Supplementary Tables 5–7. Furthermore, we performed a Pearson correlation (Supplementary Tables 8–13) based on AUC ratios of the 11 isolates in presence of all macrolide (IV, MX, EM, CH, AZ) and lincosamide (CL) compounds for each tested concentration. There, we could observe pairwise correlation within macrolide antibiotics (EM, CH, AZ) for all tested concentrations, but not between anthelminthics (IV, MX) and macrolide (EM, CH, AZ) or lincosamide (CL) antibiotics.

### Bacterial isolates display adaptation to IV/MX and a selection of antibiotics after anthelminthic challenging experiments

Since IV and MX inhibit bacterial growth in a broad range of isolates, we aimed to test whether sensitive bacterial isolates can adapt to repeated sub-inhibitory exposure of IV and MX and whether this adaptation changes the antibiotic susceptibility profiles of these isolates. Five bacterial isolates, namely *S. salivarius (01)*, *S. parasanguinis (01)*, *S. pneumoniae (02)*, *S. mitis (01)* and *S. dysgalactiae (01)*, and their challenged counterparts (appendices "-IVc" or "-MXc") were incubated with IV/MX and antibiotics as described before (Supplementary Figs. 14–18). As the primary focus of these incubations was to solely demonstrate the effect of pre-challenging on antibiotic sensitivity, we opted for sub-lethal doses (i.e., 0.1 µM or 0.01 µM) if no growth could be measured at concentrations of ≥1 µM.

Comparing the IV and MX AUC ratios of the original to the challenged isolates, we observed three phenotypes (Fig. 3). First, we documented increased AUC ratios for both IV and MX in response to anthelminthic challenging with either drug. For instance, for *S. salivarius (01-IVc)* ($AUC_{IV10µM}$: 0.73; $AUC_{MX10µM}$: 0.60) compared to *S. salivarius (01)* ($AUC_{IV10µM}$: 0.68; $AUC_{MX10µM}$: 0.51). We also noted different AUC ratios in the presence of either anthelminthic at 10 µM for *S. salivarius (01-MXc)* ($AUC_{IV10µM}$: 0.77; $AUC_{MX10µM}$: 0.91) compared to *S. salivarius (01)*. This phenotype is shared with *S. pneumoniae (02-IVc)* and *S. pneumoniae (02-MXc)*, *S. mitis (01-IVc)*, and *S. mitis (01-MXc)* and lastly *S. dysgalactiae (01-IVc)* compared to their respective unchallenged original isolate. In contrast, we observed a different phenotype in *S. parasanguinis (I01)*, namely that AUC ratios only increase when re-challenged with the same anthelminthic. For instance, *S. parasanguinis (01-IVc)* ($AUC_{IV10µM}$: 0.96; $AUC_{MX10µM}$: 0.25) possesses a higher AUC ratio in presence of IV compared to *S. parasanguinis (01)* ($AUC_{IV10µM}$: 0.79; $AUC_{MX10µM}$: 0.26), but a lower AUC ratio in presence of MX. Similarly, *S. parasanguinis (01-MXc)* ($AUC_{IV10µM}$: 0.75; $AUC_{MX10µM}$: 0.74) displays higher AUC ratios compared to *S. parasanguinis (01)* in the presence of MX, compared to lower ratios in presence of IV. Lastly, *S. dysgalactiae (01-MXc)* ($AUC_{IV10µM}$: 0.84; $AUC_{MX10µM}$: 0.19) exhibits a third and unique phenotype displaying lower AUC ratios compared to *S. dysgalactiae (01)* for both IV and MX incubations.

To compare the effects of anthelminthic-challenging on antibiotic sensitivity, we used the fold-change (FC) of AUC ratios of the original isolate divided by the AUC ratio of the challenged isolate, for each compound individually. We considered that antibiotic sensitivity was affected when (1-FC) ≥ 0.1. Following exposure to MX, we observed a change in sensitivity towards MP and IP in *S. salivarius (01)* and the highest fold-change for IP ($|1-FC_{IP1µM}| = 0.20$). These changes were not observed following exposure to IV, except for CL ($|1-FC_{CL0.01µM}| = 0.10$). For *S. parasanguinis (01-IVc)*, we observed a change in sensitivity in presence of CH, AZ and CX. For *S. parasanguinis (01-MXc)* we observed an equal pattern with the addition of CL. We observed the highest fold-change for *S. parasanguinis (01-IVc)* and *S. parasanguinis (01-MXc)* in presence of CH ($|1-FC_{CH0.01µM}| = 0.18$ and 0.21). Prolonged exposure to MX in our tested *S. pneumoniae* isolate was associated with a decrease in sensitivity to all tested antibiotics (range of (highest fold change: $|1-FC_{IP1µM}| = 0.89$)). We observed a similar result following exposure to IV, except for the antibiotic CX to which sensitivity of the challenged isolate was like that of the unchallenged isolate (highest fold change: $|1-FC_{IP1µM}| = 0.82$). In the case of *S. mitis (01-IVc)*, sensitivity towards all antibiotics except CX and TC decreased, and this decrease was the highest for MP ($|1-FC_{MP1µM}| = 0.82$) resulting in almost uninhibited growth of the challenged isolate. For *S. mitis (01-MXc)*, antibiotic sensitivity was found to be the lowest for CH ($|1-FC_{CH10µM}| = 0.78$) but did not change for EM and TC. For *S. dysgalactiae (01)*, some of the changes observed post-exposure were inverted, meaning that the unchallenged isolate was less sensitive to another antibiotic than the anthelmintic-challenged one. For instance, MX-challenging was associated with increased sensitivity to the three macrolide antibiotics (EM, CH, AZ) and CL (highest fold change: $|1-FC_{CH5µM}| = 0.75$). This was not the case for the isolate that was challenged with IV (highest fold change: $|1-FC_{TC10µM}| = 0.45$), suggesting different adaptation mechanisms in this isolate. Sensitivity was found to be higher against CX and TC but remained unchanged for MP and IP, irrespective of the anthelmintic drug used for challenging.

To complement the differences in growth dynamics established by AUC ratios, we quantified the difference in antibiotic sensitivity attributable to anthelminthic drug challenging measuring minimal inhibitory concentrations (MICs). MICs of EM, CH, AZ, CL, CX, TC, MP, and IP were established prior to and after anthelminthic challenging using Etests in duplicate. Table 3 encompasses MICs of the 5 unchallenged isolates *S. salivarius (01)*, *S. parasanguinis (01)*, *S. mitis (01)*, *S. pneumoniae (02)* and *S. dysgalactiae (01)*, as well as their IV- or MX-challenged counterparts. *S. salivarius (01)*, *S. parasanguinis (01)*, *S. pneumoniae (02)* and *S. dysgalactiae (01)* did not present vast changes regarding their MIC for all tested antibiotics. In fact, MIC's of *S. pneumoniae (02-IVc)* and *S. pneumoniae (02-MXc)* for the macrolides EM, CH and AZ seem to be lowered slightly, compared to *S. pneumoniae (02)*. Furthermore, MIC of TC for *S. dysgalactiae (01-MXc)* seems to be lower than the MIC of TC for *S. dysgalactiae (01)*. In contrast, in *S. mitis (01)* we observed significantly elevated MICs for EM (MIC *S. mitis (01)* = 0.75/1.5; MIC *S. mitis (01-IVc)* = 12/6), CH (MIC *S. mitis (01)* = 0.75/0.75; MIC *S. mitis (01-IVc)* = 3/3) and AZ (MIC *S. mitis (01)* = 2/4; MIC *S. mitis (01-IVc)* = 24/48) after IV-challenging, as well as elevated MICs for AZ (MIC

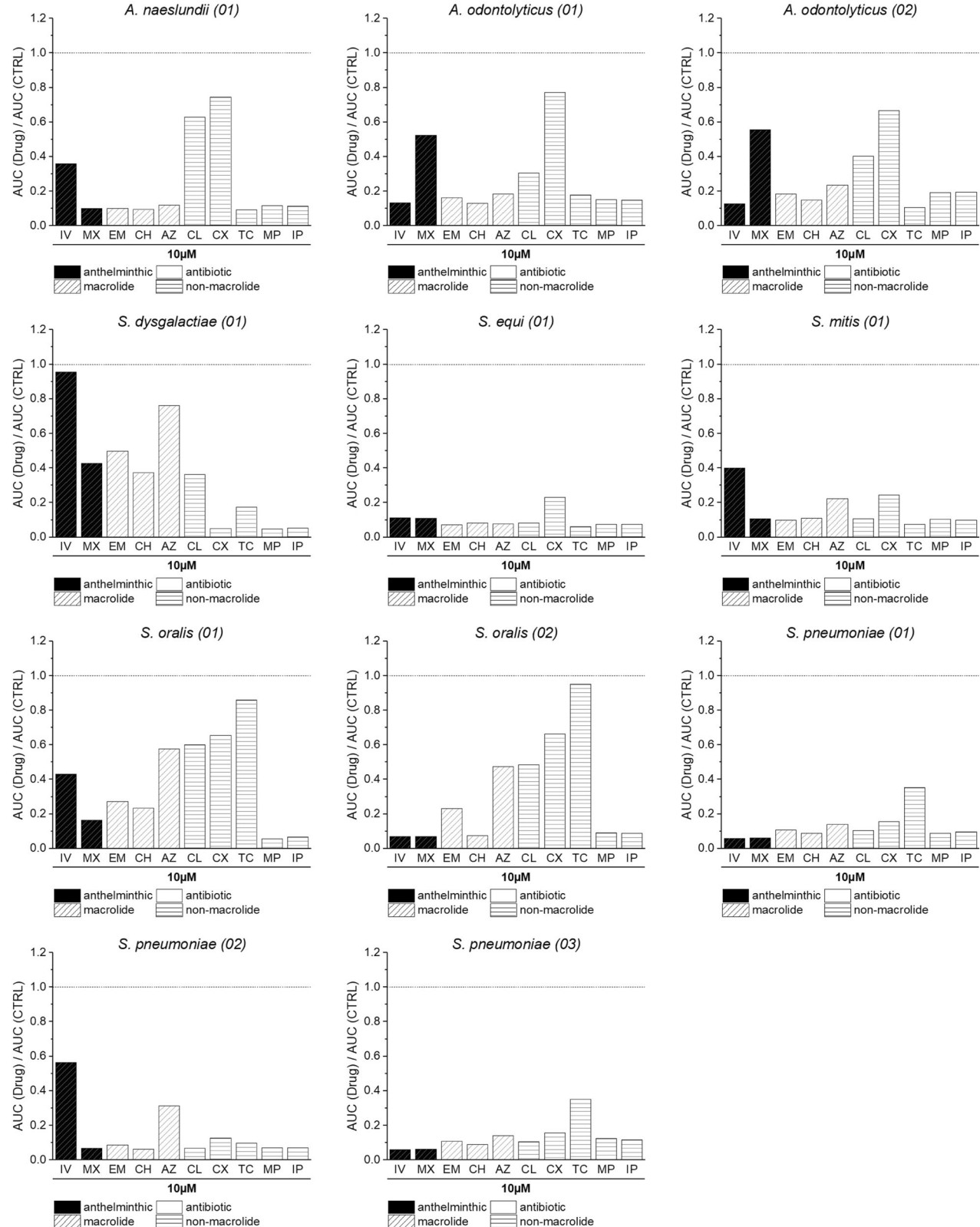

**Fig. 2 | In vitro potency of ivermectin (IV) and moxidectin (MX).** AUC ratios of 11 bacterial isolates in presence of IV, MX, EM, CH, AZ, CL, CX, TC, MP or IP at 10 μM. The dotted line marks theoretical uninhibited growth of the isolate (AUC ratio = 1). Bar colors correspond to the primary usecase of the compound. Black = anthelminthic, white = antibiotic. Bar patterns correspond to the drug class.

*S. mitis (01-MXc)* = 8/8) after MX-challenging. Lastly, we observed heterogeneous growth within the inhibition ellipse, superseding the MIC value, in case of *S. mitis (01-IVc)* with EM, CH and AZ, *S. mitis (01-MXc)* with AZ and *S. pneumoniae (02-MXc)* with CH.

## Discussion

A handful of studies have described some antibacterial properties of IV[28–30]. In this study, we systematically tested antibacterial properties of IV - and the closely related compound MX - against 59 (gut) bacterial isolates spanning

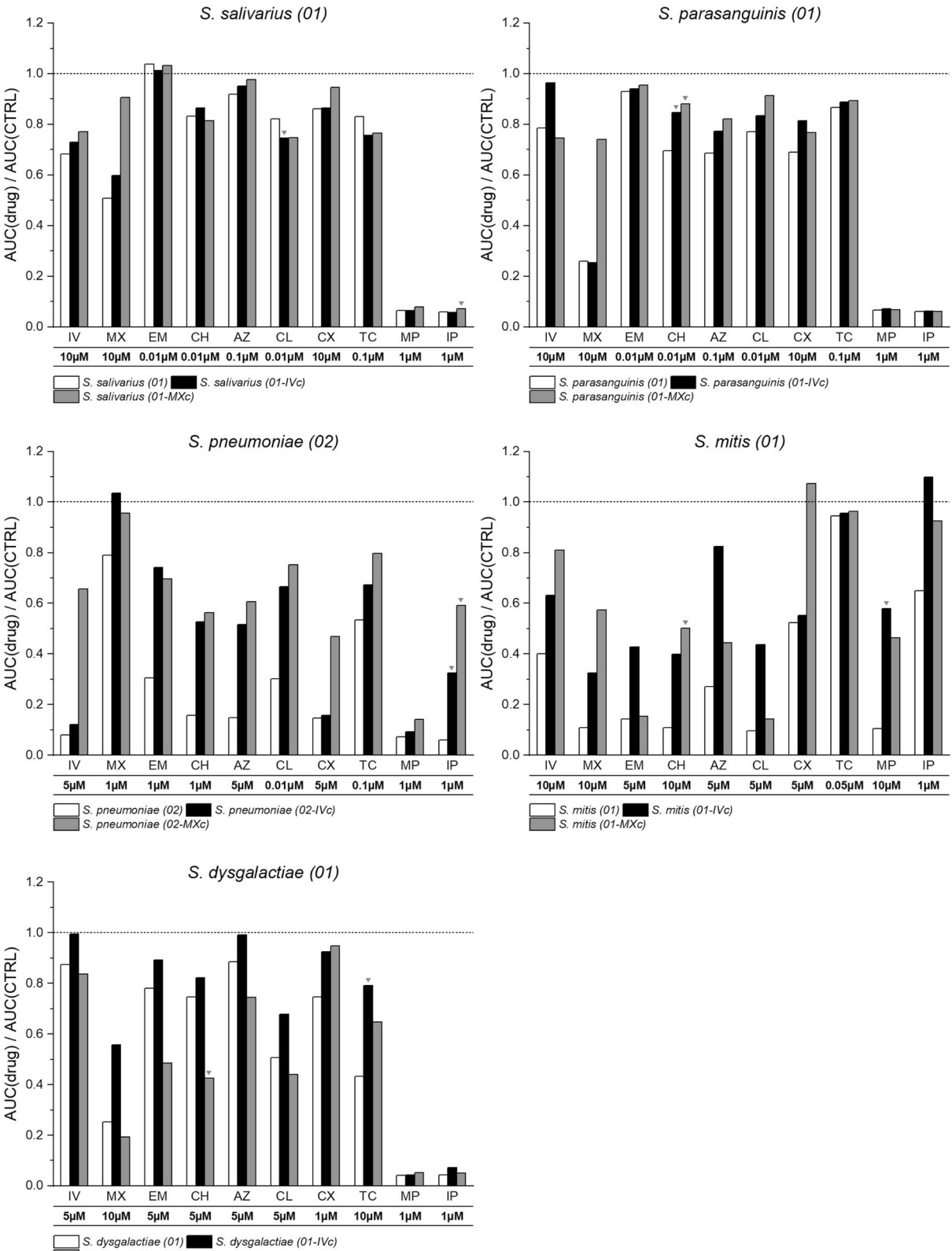

**Fig. 3 | Adaptation of bacterial isolates in response to anthelminthic challenging.** AUC ratios of 10 challenged bacterial isolates and their original counterparts in presence of IV, MX, EM, CH, AZ, CL, CX, TC, MP or IP at different concentrations. The dotted line marks theoretical uninhibited growth of the isolate (AUC ratio = 1). Bar colors correspond to the challenging status. White = original isolate, black = IV-challenged isolate, grey = MX-challenged isolate. For each challenged isolate, we highlighted the compound resulting in the highest |1-FC| (black triangles).

**Table 3 | Minimal inhibitory concentrations (MICs) of erythromycin (EM), clarithromycin (CH), azithromycin (AZ), clindamycin (CL), ciprofloxacin (CX), tetracycline (TC), meropenem (MP) and imipenem (IP) for 15 bacterial isolates**

| Isolate | MIC$_{EM}$ [ug/ml] | | MIC$_{CH}$ [ug/ml] | | MIC$_{AZ}$ [ug/ml] | | MIC$_{CL}$ [ug/ml] | | MIC$_{CX}$ [ug/ml] | | MIC$_{TC}$ [ug/ml] | | MIC$_{MP}$ [ug/ml] | | MIC$_{IP}$ [ug/ml] | |
|---|---|---|---|---|---|---|---|---|---|---|---|---|---|---|---|---|
| | R1 | R2 | R1 | R2 | R1 | R2 | R1 | R2 | R1 | R2 | R1 | R2 | R1 | R2 | R1 | R2 |
| S. salivarius (01) | 0.047 | 0.047 | 0.047 | 0.064 | 0.25 | 0.25 | 0.032 | 0.047 | 1.5 | 1.5 | 1.5 | 0.5 | 0.012 | 0.012 | 0.012 | 0.012 |
| S. salivarius (01-IVc) | 0.023 | 0.032 | 0.023 | 0.032 | 0.25 | 0.25 | 0.016 | 0.032 | 1 | 1 | 0.5 | 0.75 | 0.012 | 0.016 | 0.012 | 0.016 |
| S. salivarius (01-MXc) | 0.032 | 0.032 | 0.032 | 0.023 | 0.25 | 0.25 | 0.023 | 0.023 | 0.75 | 0.75 | 0.75 | 0.5 | 0.012 | 0.012 | 0.012 | 0.012 |
| S. parasanguinis (01) | 0.064 | 0.047 | 0.047 | 0.032 | 0.38 | 0.38 | 0.094 | 0.047 | 0.5 | 0.38 | 2 | 2 | 0.012 | 0.006 | 0.016 | 0.012 |
| S. parasanguinis (01-IVc) | 0.047 | 0.064 | 0.047 | 0.047 | 0.38 | 0.38 | 0.064 | 0.064 | 0.75 | 0.75 | 1.5 | 3 | 0.004 | 0.004 | 0.006 | 0.006 |
| S. parsanguinis (01-MXc) | 0.047 | 0.064 | 0.064 | 0.064 | 0.25 | 0.38 | 0.064 | 0.064 | 0.5 | 0.75 | 2 | 3 | 0.004 | 0.004 | 0.008 | 0.008 |
| S. mitis (01) | 0.75 | 1.5 | 0.75 | 0.75 | 2 | 4 | 0.125 | 0.094 | 2 | 2 | 0.38 | 0.38 | 1.5 | 2 | 0.38 | 0.5 |
| S. mitis (01-IVc) | 12* | 6* | 3* | 3* | 24* | 48* | 0.5 | 0.25 | 2 | 2 | 0.5 | 0.38 | 1.5 | 2 | 0.38 | 0.38 |
| S. mitis (01-MXc) | 1.5 | 1.5 | 1.5 | 1 | 8* | 8* | 0.047 | 0.047 | 3 | 2 | 0.5 | 0.38 | 1.5 | 1.5 | 0.38 | 0.38 |
| S. pneumoniae (02) | 8 | 8 | 12 | 12 | 48 | 48 | 0.047 | 0.094 | 0.5 | 0.5 | 0.38 | 0.25 | 0.008 | 0.006 | 0.004 | 0.004 |
| S. pneumoniae (02-IVc) | 6 | 6 | 8 | 8 | 24 | 24 | 0.032 | 0.032 | 0.38 | 0.5 | 0.19 | 0.25 | 0.008 | 0.008 | 0.004 | 0.006 |
| S. pneumoniae (02-MXc) | 8 | 8 | 8* | 6* | 64 | 48 | 0.047 | 0.094 | 0.75 | 0.75 | 0.19 | 0.19 | 0.008 | 0.006 | 0.004 | 0.004 |
| S. dysgalactiae (01) | >256 | >256 | >256 | >256 | >256 | >256 | >256 | >256 | 0.25 | 0.38 | 12 | 16 | 0.012 | 0.012 | 0.006 | 0.006 |
| S. dysgalactiae (01-IVc) | >256 | >256 | >256 | >256 | >256 | >256 | >256 | >256 | 0.38 | 0.5 | 12 | 12 | 0.012 | 0.012 | 0.006 | 0.006 |
| S. dysgalactiae (01-MXc) | >256 | >256 | >256 | >256 | >256 | >256 | >256 | >256 | 0.38 | 0.38 | 3 | 6 | 0.012 | 0.023 | 0.008 | 0.008 |

For MIC's marked with an asterisk (*), we found considerable, yet heterogeneous growth within the inhibition zone superseding the MIC value.

*IVc* ivermectin-challenged, *MXc* moxidectin-challenged, *R1* replicate 1, *R2* replicate 2.

across 10 genera and originating from both clinical and commercial sources. We included lincosamide and macrolide-resistant clinical isolates (20/59) to test whether macrolide or lincosamide resistance is coupled to sensitivity to IV/MX. We further included clinical isolates (27/59) enriched from stool samples obtained in a study in Lao PDR, due to their extensive clinical background. Lastly, we included several commercial strains (12/59) to broaden the taxonomic range of our results. We found that both IV and MX inhibit growth of a broad range of bacterial isolates in vitro in a dose-dependent manner with comparable potency. Macrolide and lincosamide antibiotics achieve their bacteriostatic or bactericidal activity against gram-positives by binding to the 50S subunit of the bacterial ribosome. For IV and MX no MOA against bacteria has been identified yet. Given our data however, we can speculate that the MOA of IV and MX seems to be different from macrolide or lincosamide antibiotics, as, first, we observed different growth dynamics of bacterial isolates in response to the macrolides EM, CH and AZ and the lincosamide CL compared to IV and MX (Supplementary Tables 8–13). Secondly, we observed changes in AUC ratios of the macrolide or lincosamide-resistant isolates S. pneumoniae (02), S. dysgalactiae (01) and S. mitis (01) after anthelminthic challenging, indicating a further adaptation (different MOA), opposed to a redundant one (same MOA).

We demonstrated adaptation to IV and MX, associated with decreased sensitivity in vitro, assessed by AUC ratios. A possible shortcoming of our study is that aggressive in vitro challenging might not be comparable to real-world applications of IV and MX. To control STH infections, mass drug administration (MDA) is usually conducted once to twice a year. According to Maier et al., 200 µg/kg IV (i.e., the standard dose used for helminth infections) result in an estimated intestinal concentrations of 4.6 µM[21]. However, IV was recently considered as a vector control drug against female *Anopheles spp.* mosquitos, potentially introducing more frequent and higher doses[25,31]. Furthermore, the recent MORDOR trial demonstrated increased macrolide resistance within the gut microbiome already after administration of the macrolide antibiotic AZ twice a year for 2 years[32,33].

Assessed by AUC ratios, we observed broad adaptation phenotypes for S. salivarius (01), S. pneumoniae (02) and S. mitis (01), as challenging with anthelminthics resulted in decreased sensitivity against both anthelminthics. In contrast, anthelminthic challenging of S. parasanguinis (01) resulted in decreased sensitivity only against the corresponding anthelminthic, suggesting a drug-specific mechanism, opposed to a potentially broad

mechanism as adopted by S. salivarius (01), S. pneumoniae (02) and S. mitis (01). Interestingly, in the macrolide and lincosamide-resistant isolate S. pneumoniae (02), anthelminthic challenging seemed to exert a broad-spectrum effect as both, IV- and MX-challenged isolates displayed decreased sensitivity towards all antibiotic classes it was tested against (macrolides, lincosamide, tetracycline, fluoroquinolone, and carbapenem). In addition, this broad-spectrum interaction between anthelminthic and macrolide or lincosamide resistance was also observed in S. mitis (01-IVc), S. mitis (01-MXc) and S. dysgalactiae (01-IVc). The broad adaptation phenotypes and differences in growth dynamics observed by AUC ratios are only partially reflected in MIC values. In fact, we did not observe noteworthy differences in MICs for S. salivarius (01), S. parasanguinis (01), S. pneumoniae (02) and S. dysgalactiae (01) prior to and after anthelminthic challenging. Interestingly, we observed substantial cross-reactivity of IV-challenging and sensitivity to macrolides and a link between MX-challenging and sensitivity to AZ in S. mitis (01). Noteworthy, S. mitis (01-IVc) reached the clinical MIC breakpoint of 0.5 µg/ml for CL set by the European Committee on Antimicrobial Susceptibility Testing (EUCAST) in January 2024 in one of two replicated Etests. While analysis of the AUC ratios would suggest broad phenotypes of decreased antibiotic sensitivity across isolates, MIC values suggest limited cross-reactivity of anthelminthics, with the exception of S. mitis (01), which presents substantial cross-reactivity geared towards the macrolide antibiotics EM, CH and AZ. Importantly, AUC ratios are solely based on growth dynamics, which do not inherently influence MICs. Importantly, changes in growth dynamics may already shift the compositional balance in the gut microbiome – given that this environment is highly competitive – with noteworthy consequences. In fact, Schneeberger et al. recently demonstrated the association between pre-treatment gut bacterial composition and treatment outcome of ALB-IV combination therapy against STH infections[22]. Nonetheless, further characterization of resistance mechanisms is crucial to understand the development and maintenance of AMR in clinically relevant species. This is specifically true for S. pneumoniae which is one of the six leading pathogens causing lethal lower respiratory infections[34]. S. pneumoniae is primarily found in the upper respiratory tract, therefore, in its primary niche, contact with high concentrations of IV or MX is unlikely. However, dynamic transfer of resistance genes across the microbiome was recently demonstrated[35] and would a possible scenario in the case of IV/MX MDA. It is also critical to investigate this interaction in other commensal

species as they might act as reservoir for AMR genes and thus contribute to the persistence of AMR in the human body. Populations at risk for helminth infections and other infectious diseases often overlap in the Global South. For instance, most deaths related to AMR are currently observed in sub-Saharan Africa[34]. There, anthelminthics could further contribute to the burden of AMR in bacteria. Additionally, both Southeast Asia and India represent hotspots of helminth prevalence[36] and antibiotic use[37] thus providing a permissive environment for the described dynamics between anthelminthics and antibiotic-resistant bacteria. Therefore, the observed effect of anthelminthic challenging on antibiotic-resistant bacteria should be investigated in further studies.

Our study presents few limitations. Firstly, it does not uncover the mode of action of IV and MX in bacteria and corresponding adaptation mechanisms on a molecular level. In further studies, we aim to conduct genomic and transcriptomic analyses to fill this gap. Furthermore, these molecular analyses should be extended to human cohort samples to detect whether off-target effects of the two anthelminthics actually manifest in a clinical setting. Moreover, our experiments did not test for long-term adaptation. Thus, the described adaptation could be transient. This shortcoming should also be addressed in further experiments. In conclusion, our study yielded three key novel findings: (i) exposure to IV and MX causes phenotypical adaptation, (ii) adaptation mechanisms are likely diverse, encompassing both drug-specific and broad-spectrum mechanisms, and (iii) adaptation to IV/MX also imparts varying degrees of decreased sensitivity to other antibiotics. Both IV and MX remain crucial compounds to combat several infectious diseases. However, in the context of the uprising of IV and MX, this raises serious concerns, as repeated anthelminthic exposure of gut-bacterial isolates may trigger broad cross-resistant phenotypes, potentially interfering with vital antibiotic treatments.

## Methods

### Origin of bacterial isolates

20/59 bacterial isolates are macrolide or lincosamide-resistant and were obtained via the Institute for Infectious Diseases (Berne, Switzerland). As lincosamides share the mode of action with macrolides, resistance mechanisms often overlap[38]. We therefore included these isolates to test whether macrolide or lincosamide resistance is further coupled to sensitivity to IV/MX. The lincosamide-resistant strains were comprised of 5 *Streptococcus spp.* and *3 Actinomyces spp. Streptococcus* species are frequent, gram-positive commensals in the oral microbiota, but are especially abundant in the small intestine[39]. However, several mechanisms of macrolide resistance have been identified among *Streptococcus* species, including mainly drug efflux, target alteration, and drug inactivation[40–42]. Moreover, *S. salivarius* – amongst others – was associated with ALB-IV combination treatment failure in a recent study[22]. Building on this concept, we expanded the number of *Streptococcus* species in our study to test for individual interactions. *Actinomyces* spp. on the other hand are close relatives to avermectin-producing bacteria (*Streptomyces avermitilis*). They therefore likely possessed unique co-incubation phenotypes with derivatives such as IV or MX. The macrolide-resistant isolates were comprised solely of *S. pneumoniae* isolates ($n = 12$). *S. pneumoniae* is primarily found in the lungs but also commonly found throughout the upper and lower GI tract, and is one of the six leading pathogens causing lethal lower respiratory infections[34]. We included 27/59 bacterial isolates from clinical stool samples from a recent study conducted in Lao PDR (NCT03527732). As Lao PDR represents a prime site of clinical trials involving both IV and MX and lies in midst of a hotspot of helminth prevalence, the corresponding bacterial isolates likely reflect an extensive clinical background. Bacterial isolation from stool samples was either conducted at Swiss TPH in Allschwil, Switzerland (18/59 bacterial isolates) or by Dr. Julian Garneau and Alison Gandelin at the University of Lausanne in Lausanne, Switzerland (9/59 bacterial isolates). Enriched clinical isolates were comprised of the genera *Blautia, Clostridium, Dorea, Enterococcus, Escherichia* and *Streptococcus*. It is critical to investigate this interaction with IV/MX in several commensal species as they might act as reservoir for AMR genes and thus contribute to the persistence of AMR in

the human body. Therefore, we included 12/59 commercial isolates (*Bacteroides, Blautia, Dorea, Lactobacillus, Staphylococcus, Streptococcus*), to further broaden taxonomic range of our results. The commercial isolates were purchased from the German Collection of Microorganisms and Cell Cultures (https://www.dsmz.de/). We additionally generated 10 anthelminthic-challenged isolates through a procedure described later. We opted for *Streptococcus* species due to reasons mentioned above. Moreover, we included lincosamide-resistant (*S. mitis, S. dysgalactiae*), macrolide resistant (*S. pneumoniae*) and commercial strains (*S. salivarius, S. parasanguinis*) to characterize the phenotype of anthelminthic challenging in these differing genotypes. An overview of the challenged bacterial isolates is given in Supplementary Table 4.

### Enrichment and identification of clinical isolates

Clinical stool samples were obtained from a recent study in Lao PDR (NCT03527732)[40]. For each sample, 50–100 mg of stool were homogenized in 500 μl BHI + 5% yeast in a safety cabinet (aerobic conditions). After centrifugation (1 min, 15,000 rcf), 80 μl of the supernatant was added to 10 ml of the corresponding enrichment medium (Supplementary Table 2) and left overnight in a 5% $CO_2$ incubator at 37 °C. The following culture media were used: Brain heart infusion broth (BHI; Thermo Fisher Scientific, CM1135), yeast extract (Thermo Fisher Scientific, LP0021B), inulin (Thermo Fisher Scientific, 457100250), dehydrated Todd-Hewitt broth (TH; Thermo Fisher Scientific, CM0189), modified gifu anaerobe medium (mGAM; HyServe, 05433), dehydrated Schaedler broth (Thermo Fisher Scientific, CM0497B) and Lennox broth (LB; Thermo Fisher Scientific, 12780052). The next day, each culture was diluted 1:1000 in BHI + 5% yeast and 10 μl were streaked on a BHI + 5% yeast agar (Thermo Fisher Scientific, CM1136) plate. The streaked agar plates were left overnight in a 5% $CO_2$ incubator at 37 °C. The following day, individual colonies were randomly picked and re-inoculated in fresh BHI + 5% yeast. Again, cultures were left to grow overnight in a 5% $CO_2$ incubator at 37 °C. Finally, 20% glycerol stocks were prepared of each isolate and stored at –80 °C. To identify the isolated bacteria, Columbia agar plates with 5% sheep blood (Thermo Fisher Scientific, PB5039A) were streaked with the corresponding glycerol stocks. Colonies were left to grow overnight in a 5% $CO_2$ incubator at 37 °C. The next day, MALDI-TOF analysis was performed by the Institute of Medical Microbiology (University of Zürich, Switzerland) to identify the isolates. In brief, single colonies are loaded onto a MALDI-TOF steel target plate using a sterile toothpick. Each colony on the plate is treated with 1 μl of 25% formic acid and subsequently with 1 μl of α-Cyano-4-hydroxycinnamic (CHAC) matrix. After sample preparation, the plate is loaded into a Microflex MALDI-TOF instrument for spectra measurement (Bruker Daltonics) and subsequent identification[43].

### Cultivation of bacterial isolates

Bacterial glycerol stocks were stored at –80 °C. To cultivate an isolate, the glycerol stock was first thawed on ice. Subsequently, 10 μl of the thawed glycerol stock was used to start a culture in 10 ml of culture medium and left to grow at 37 °C. BHI + 5% yeast was used exclusively to cultivate all bacterial isolates. The incubation period for all isolates was between 24 h and 72 h. Anaerobic cultivation was performed in a vinyl anaerobic chamber (Coy Laboratory Products, Michigan, United States) with a gas mix composed of 85% $N_2$, 10% $CO_2$, and 5% $H_2$. Inoculated isolates were grown in the integrated anaerobic incubator at 37 °C. Aerobic inoculations were performed in a safety cabinet (SKAN Berner, Elmshorn, Germany). Inoculated aerobic isolates were grown in a 5% $CO_2$ incubator (Binder, Tuttlingen, Germany) at 37 °C. To preserve bacterial cultures after cultivation, 100 μl of culture was added to 100 μl of 40% glycerol solution (Thermo Fisher Scientific, 17904) in purified water. The resulting 20% bacterial glycerol stocks were frozen and stored at –80 °C until further use.

### Bacterial anthelminthic challenging experiments

Five bacterial isolates - originally *S. salivarius (01), S. parasanguinis (01), S. pneumoniae (02), S. mitis (01)* and *S. dysgalactiae (01)* - were challenged with

increasing concentrations of anthelminthics (IV, MX) up to a final concentration of 20 µM – or 10 µM in the case of *S. pneumoniae (02-MXc) and S. mitis (01-MXc)* -, resulting in IV or MX challenged isolates *S. salivarius (01-IVc), S. salivarius (01-MXc), S. parasanguinis (01-IVc), S. parasanguinis (01-MXc), S. pneumoniae (02-IVc), S. pneumoniae (02-MXc), S. mitis (01-IVc), S. mitis (01-MXc), S. dysgalactiae (01-IVc)* and *S. dysgalactiae (01-MXc)* (Supplementary Table 4 and Supplementary Fig. 1). We ordered both IV (I8898) and MX (PHR1827) from Merck & Cie (Buchs, Switzerland).

In brief, each isolate was initially cultured from a glycerol stock by adding 10 µl of stock to 10 ml growth medium (BHI + 5% Yeast) and grown for 24 h in a 5% $CO_2$ incubator at 37 °C. After 24 h, the bacterial isolates entered the 5 µM challenging iteration: A volume of B1 (=bottleneck at 5 µM) of the saturated culture was re-cultured in 10 ml growth medium supplemented with 5 µM IV or MX and left to grow for a duration of G1 (=generation time at 5 µM) in a 5% $CO_2$ incubator at 37 °C. Initially, we set G1 to 24 h for all isolates. (1) If an OD600 > 0.8 was not reached after 24 h, we extended G1 to 48 h. (2) If an OD600 > 0.8 was not reached after G1 = 48 h, we re-cultured a volume of B1 of this bacterial suspension in 10 ml growth medium supplemented with 5 µM IV or MX and incubated the new culture for a duration of G1 = 48 h again. As a result P1 (=number of passages at 5 µM) of that culture was increased to 2. This step was repeated a maximum of two times. If there culture still did not reach saturation after three passages (P1 = 3), the challenging in that condition was aborted. (3) If an OD600 > 0.8 was reached after G1 = 24 h or 48 h and P1 ≤ 3, this culture entered the 10 µM challenging iteration.

All isolates that successfully completed the the 5 µM challenging iteration, entered the 10 µM challenging iteration: A volume of B2 (=bottleneck at 10 µM) of the saturated culture—that was grown in 5 µM IV or MX -, was re-cultured in 10 ml growth medium supplemented with 10 µM IV or MX and left to grow for a duration of G2 (=generation time at 10 µM) in a 5% $CO_2$ incubator at 37 °C. Initially, we set G2 to 24 h for all isolates. (1) If an OD600 > 0.8 was not reached after 24 h, we extended G2 to 48 h. (2) If an OD600 > 0.8 was not reached after G2 = 48 h, we re-cultured a volume of B2 of this bacterial suspension in 10 ml growth medium supplemented with 10 µM IV or MX and incubated the new culture for a duration of G2 = 48 h again. As a result, P2 (=number of passages at 10 µM) of that culture was increased to 2. This step was repeated a maximum of two times. If there culture still did not reach saturation after three passages (P2 = 3), the challenging in that condition was aborted. (3) If an OD600 > 0.8 was reached after G2 = 24 h or 48 h and P2 ≤ 3, this culture entered the 20 µM challenging iteration. All isolates that successfully completed the 10 µM challenging iteration, entered the 20 µM challenging iteration: A volume of B3 (=bottleneck at 20 µM) of the saturated culture – that was grown in 10 µM IV or MX -, was re-cultured in 10 ml growth medium supplemented with 20 µM IV or MX and left to grow for a duration of G3 (=generation time at 20 µM) in a 5% $CO_2$ incubator at 37 °C. Initially, we set G3 to 24 h for all isolates. (1) If an OD600 > 0.8 was not reached after 24 h, we extended G3 to 48 h. (2) If an OD600 > 0.8 was not reached after G3 = 48 h, we re-cultured a volume of B3 of this bacterial suspension in 10 ml growth medium supplemented with 20 µM IV or MX and incubated the new culture for a duration of G3 = 48 h again. As a result P3 (=number of passages at 20 µM) of that culture was increased to 2. This step was repeated a maximum of two times. If there culture still did not reach saturation after three passages (P2 = 3), the challenging in that condition was aborted. (3) Challenging at 20 µM was considered successful if an OD600 > 0.8 was reached after G3 = 24 h or 48 h and P3 ≤ 3.

We prepared 20% glycerol stocks for each culture in each iteration of the challenging experiment and stored them at –80 °C until further use. To confirm identity and exclude contamination, potentially introduced by long incubation periods and multiple passages, the final challenged isolates underwent 16S rRNA gene sequencing on the ONT MinION platform using the ONT Native Barcoding Kit (SQK-NBD114.24) on a Flongle Flow Cell (FLO-FLG114) according to the manufacturer's instructions. The resulting sequences were identified using Emu[44] equipped with its full-length 16S database.

## MIC determination before and after challenging experiment via Etest

MICs of *S. salivarius (01)*, *S. parasanguinis (01)*, *S. pneumoniae (02)*, *S. mitis (01)*, *S. dysgalactiae (01)* and their anthelminthic challenged counterparts *S. salivarius (01-IVc)*, *S. salivarius (01-MXc)*, *S. parasanguinis (01-IVc)*, *S. parasanguinis (01-MXc)*, *S. pneumoniae (02-IVc)*, *S. pneumoniae (02-MXc)*, *S. mitis (01-IVc)*, *S. mitis (01-MXc)*, *S. dysgalactiae (01-IVc)* and *S. dysgalactiae (01-MXc)* were determined via Etests. We ordered Etests for EM (412334), CH (412313), AZ (412257), CL (412315), CX (412311), TC (412471), MP (412402) and IP (412374) from bioMérieux Suisse S.A, Geneva, Switzerland. In brief, 10 µl of glycerol stock was cross-streaked on a BHI + 5% Yeast agar plate. In the case of IV-challenged isolates, the agar plates additionally contained 20 µM IV. Vice versa, MX-challenged isolates were cultivated on agar plates imitating the former challenging environment (addition of 10 µM MX in the case of *S. pneumoniae (02-MXc)* and *S. mitis (01-MXc)*; addition of 20 µM MX in all other cases). The agar plates were incubated in a 5% $CO_2$ incubator at 37 °C for 24 h. Subsequently and for each plate 3 single colonies were picked and homogenized in 200 µl saline solution (Prompt Inoculation System, Beckman Coulter, B1026-10D) in duplicate. The homogenized solution for each isolate and duplicate was streaked on a Mueller-Hinton + 5% sheep blood agar plate (Thermo Fisher Scientific, PB5007A) with a sterile cotton swab. Subsequently, Etests were applied using sterile tweezers. The agar plates were incubated in a 5% $CO_2$ incubator at 37 °C for 18 h. After 18 h, we imaged each plate. Blinded MIC readout was performed by a trained co-worker (Carlo Casanova, Institute for Infectious Diseases IFIK, Berne, Switzerland).

## In vitro incubation assays with anthelminthic or antibiotic drugs

Throughout this study, bacterial isolates were incubated with a variety of anthelminthics (IV, MX) or antibiotics, namely EM (Lubio, ORB322468), CH (Lubio, HY-17508), AZ (Lubio, ORB322360), CL (Lubio, HY-B0408A), TC (Lubio, HY-B0474), CX (Lubio, HY-B0356A), MP (Lubio, A5124) and IP (Lubio, ORB1308410). Incubations were carried out in a 96-well plate format to facilitate optical density measurements (at $\lambda = 600$ nm; OD600) using a Hidex Sense plate reader (Hidex Oy, Turku, Finland). OD600 was measured every 30–60 min at 37 °C for 16–72 h. Drug solutions were prepared as follows: First, the corresponding drug amounts were weighed ±0.1 mg on an Ohaus Adventurer AX124 analytical scale (Ohaus, Nänikon, Switzerland) and dissolved in the according volume of solvent to generate a 5000 µM working solution (WS). We used UltraPure™ DNase/RNase-free distilled water (Thermo Fisher Scientific, 10977-035) to dissolve CL, CX, TC and IP. Dimethylsulfoxide (DMSO; Sigma-Aldrich, 41640) was used to dissolve IV, MX, EM, CH, AZ and MP. 2500 µM and 500 µM WS were obtained by diluting the 5000 µM WS. Working solutions were prepared fresh each week and kept at –20 °C while limiting the amount of freeze and thaw cycles. WS only represent an intermediate step and had to be diluted in BHI + 5% yeast to 1.25× of the desired final assay concentration (12.5 µM, 6.25 µM, 1.25 µM). Prior to the incubation, the bacterial isolates were cultivated from glycerol stocks as described before. In the case of challenged isolates, 20 µM of either IV or MX - or 10 µM in the case of *S. pneumoniae (02-MXc)* and *S. mitis (01-MXc)* - was added to the culture medium, to replicate the former cultivation environment. Grown bacterial cultures were diluted 1:200 in BHI + 5% yeast, before combining them with the corresponding 1.25× drug solutions. In the case of challenged isolate cultures, the dilution was 1:20,000. In this case, we can not exclude that traces IV or MX are taken over to the incubation assay, but given the dilution factor we consider this negligible. Each well of a plate contained a ratio of bacterial culture: 1.25× drug solution of 1:4. The final 1× drug concentration was therefore reached and DMSO concentration was constant at 0.2%. During each incubation, we added two control per isolate: As some drugs needed to be dissolved in DMSO – which is known to inhibit bacterial growth in higher concentrations[45] - a

DMSO control (isolate in 0.2% DMSO) was added to each plate. A second control represented a positive control (PC), meaning the isolate was added to growth medium only. We incubated 28/59 isolates under anaerobic conditions as they either did not tolerate oxygen (strict anaerobes) or were enriched under anaerobic conditions. Likewise, 29/59 isolates were incubated under aerobic conditions. The remaining 2/59 isolates (*S. salivarius, S. parasanguinis*) we incubated in both conditions, to test whether the resulting phenotypes would differ.

## Bacterial DNA extraction
To extract bacterial DNA from liquid cultures, we used the DNeasy PowerSoil Pro kit (QIAGEN, 47016) and followed the manufacturer's standard protocol included in the kit using 100 μl of the bacterial culture. Final DNA concentrations were determined using a QuBit™ 4 fluorometer (Thermo Fisher Scientific, Q33238) with the 1× dsDNA HS Assay kit (Thermo Fisher Scientific, Q33231). Eluates were stored at −20 °C until further use.

## Library preparation for bacterial 16S rRNA sequencing
Bacterial 16S rRNA sequencing was performed on the MinION Mk1C platform (Oxford Nanopore Technologies, United Kingdom of Great Britain). All samples were processed according to the manufacturer's instructions provided with the 1-24 16S barcoding kit (Oxford Nanopore Technologies, SQK-16S024). In brief, the DNA concentration of all samples was measured using a QuBit™ 4 fluorometer (Thermo Fisher Scientific, Q33238) and the 1× dsDNA HS Assay kit (Thermo Fisher Scientific, Q33231). Samples were diluted using UltraPure™ DNase/RNase-free distilled water (Thermo Fisher Scientific, 10977-035) to reach a final input of 25 ng DNA. Next, the samples were barcoded and amplified on a CFX Opus 384 Real-Time PCR system (Biorad, Cressier, Switzerland). Subsequently, the barcoded amplicons were cleaned up in multiple steps using the AMPure XP reagent (Beckman Coulter, A63881) in combination with the DynaMag™-2 magnetic tube stand (Thermo Fisher Scientific, 12321D) and a HulaMixer™ (Thermo Fisher Scientific, 15920D). Finally, the sequencing libraries were pooled and sequenced using a flongle flow cell (Oxford Nanopore Technologies, FLG-001). Sequencing was performed for ~24 h.

## Agarose gel electrophoresis
After the amplification step within the 1–24 16S barcoding kit protocol, gel electrophoresis was performed to visualize successful 16S rRNA gene amplification. A 1% agarose solution in 1× Tris-acetate-EDTA (Thermo Fisher Scientific, B49) was prepared using analytical grade agarose powder (Promega, V3121). The GeneRuler 1 kb Plus DNA Ladder (Thermo Fisher Scientific, SM1331) and samples were combined with 6× DNA Loading Dye (Thermo Fisher Scientific, R0611) in a 5:1 ratio (sample:dye). Gel electrophoresis was running in 1× TAE for 25 min at 100 V. For final staining, the gel was bathed in a GelRed® nucleic acid gel stain solution (Biotium, 41003) for 20 min. UV imaging was performed using a Lourmat Quantum ST5 Imager (Vilber, Collégien, France). Images were taken using the VisionCapt software (v15.0) with an exposure time of 2–3 s.

## Data analysis
Data of bacterial growth curves, as well as AUC ratios were curated in Microsoft Excel 2016 and plotted using OriginPro 2022b (OriginLab Corporation, Northhampton MA, United States). To calculate AUC ratios the R package growthcurver[42] (v0.3.1) was used in RStudio equipped with R 4.1.3. All original growth curves that were used for AUC ratio calculation can be found in the Supplementary Material.

## Statistics and reproducibility
To test for dose dependence in IV/MX incubations, we applied a pairwise Wilcoxon rank sum exact test (to test AUC ratio pairs for two

concentrations; *p*-value adjustment method = BH, two-sided) and a kruskal wallis test (to test corresponding AUC ratios across three concentrations; degrees of freedom = 2) in RStudio equipped with R 4.1.3 to the averaged duplicate AUC ratio values of IV/MX-sensitive isolates (Tables 1–2). To test, for each incubation concentration, whether potency of the antheminthics is comparable to antibiotics, we utilized a Kruskal–Wallis test (degrees of freedom = 1) with the averaged duplicate data of 11 bacterial isolates (Fig. 2, Supplementary Figs. 12–13). *P*-values for pairwise comparisons of anthelminthic and antibiotic compounds (Supplementary Tables 5–7) were obtained via a Wilcoxon rank sum exact test (*p*-value adjustment method = BH, two-sided). The Pearson correlation matrix and *p*-values for averaged duplicate AUC ratio values of 11 bacterial isolates in presence of IV/MX or macrolides/lincosamide antibiotics were calculated in Microsoft Excel 2016.

## Data availability
The sequencing data (16S rRNA gene sequencing) generated in this study have been deposited in the NCBI Short Read Archive under the accession PRJNA1053597. Numerical source data underlying all figures have been deposited under https://doi.org/10.6084/m9.figshare.27248703[46].

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

## Acknowledgements

We thank the Lao TPHI team, local authority, field research team, and study participants for facilitating the clinical trial NCT03527732. We are thankful to Diana Albertos Torres (Institute of Medical Microbiology, University of Zürich, Switzerland), who kindly agreed to identify 18 bacterial colonies isolated from stool by MALDI-TOF-MS. We are grateful to the European Research Council (No. 101019223) for financial support.

## Author contributions

J.D.: study design, research design, project supervision, experimental work, statistical analyses, figure generation, writing of the initial paper and paper editing. J.K.: study design, research design, project supervision, funding acquisition and paper editing. S.S: study design, research design, project supervision, field implementation. J.G. and A.G.: experimental work (enrichment of clinical isolates), paper editing. C.C. and P.M.K.: paper editing. P.V.: project supervision (enrichment of clinical isolates) and paper editing. P.H.H.S.: study design, research design, project supervision and paper editing.

## Competing interests

The authors declare no competing interests.
