## [Transparent Peer Review file · Communications Biology]

Exposure of gut bacterial isolates to the anthelmintic drugs, ivermectin and moxidectin, leads to antibiotic-like phenotypes of growth inhibition and adaptation.

Corresponding Author: Dr Pierre Schneeberger

Version 0:

Reviewer comments:

Reviewer #1

(Remarks to the Author)

In this study, the authors explored the antibacterial activity of two anthelmintic drugs (ivermectin and moxidectin) on a diverse set of 59 bacterial isolates. They compared their activity with 8 commonly used antibiotics and show that challenging some of these strains with the anthelmintic drugs can have an impact on antibiotic sensitivity. Overall the concept of the study is timely as the impact of non-antibiotics on diverse bacterial species starts to be unravelled and the potential cross-resistance between non-antibiotics and antibiotics is an emerging concept. However, the authors should re-consider some of the methodology they use to fit with what is generally done in the field. At the moment, very few conclusions can be taken from the data, which makes the study very descriptive. See below:

Major comments:

- These two drugs were part of the Maier et al., Nature, 2018 study, where they showed they had an activity on some gut microbes. In that study, the concentration used was 20 μM and they only tested 40 gut species. Therefore the idea that these two drugs affect some bacterial species is not new, but here the authors used several drug concentration and another set of bacterial species.
- My major concern is the method to measure drug sensitivity/resistance based on AUC ratio. The whole antibiotics field typically uses the measure of MIC or IC₉₀ to compare drug activities and call for sensitivity or resistance. Here, the authors only use the AUC ratio, which does not allow them to actually compare sensitivity or resistance between strains. Measuring MICs would allow to get quantitative data that can actually be compared.
- The challenging experiment is not clearly described in the text. How many passages have been done, for how many generation and at which drug concentration? How many parallel lineages are tested for each condition (and do they exhibit the same changes)? What is the bottleneck or dilution at each passage? Is resistance to anthelmintics actually reached (i.e. increased MIC after the challenge). My overall perception is that it is not clear whether what the authors describe comes from the cells affected from the first treatment and therefore becoming more "sensitive" or "resistant" to the second drug (drug combination effect) or if really the real resistance (increased MIC) to the first drugs make them more sensitive/resistant (collateral sensitivity or cross-resistance).

Minor comment:

- The rationale for choosing the 59 strains is not entirely clear to me.

Reviewer #2

(Remarks to the Author)

The authors designed an in vitro experiment aiming to test the antibiotic sensitivity of several gut bacteria after exposure to ivermectin and moxidectin. However, the innovation and significance are limited. Thus, I don't think this manuscript is suitable for publication in *Communications Biology*.

Reviewer #3

(Remarks to the Author)

The study focuses on the antibiotic properties of two widely used anthelmintic medications and how they might affect the emergence of cross-resistance to macrolides and other antibiotic families, against various bacterial isolates. Such a research topic is timely and relevant; however, I have a few comments to make:

Overall, it is correct that ivermectin and moxidectin are closely related broad-spectrum anthelmintics, both produced through fermentation by soil actinomycetes (*Streptomyces*). They also share some structural and physicochemical properties and a common mode of action. Therefore, considering the location of the target organism, we expect these two anthelmintic drugs to exhibit off-target effects and, to some extent, cross-react with other antibiotics. Additionally, perturbation of gut microbial growth is a complex process influenced by various factors including diet.

More specific comment:

1. Lines 55–57: Different settings have recorded varying responses to different anthelmintics. I suggest you include all human STHs with reported poor cure rates, as this study does not focus on a particular STH species.

Version 1:

Reviewer comments:

Reviewer #1

(Remarks to the Author)

The authors properly addressed my concerns, especially regarding the MIC data.

Reviewer #3

(Remarks to the Author)

Authors have addressed all comments raised in the previous version, except that supplementary Table 4 is missing from the revised paper.

Point-by-point response:

We thank the editor and the reviewers for providing the opportunity to revise and resubmit our manuscript. We are confident that the overall quality of our manuscript has further improved by addressing the suggestions put forth by all referees. Hence, we hope that it will now be suitable to be considered for publication in *Communications Biology*.

Our responses are highlighted in blue.

Reviewer #1 (Remarks to the Author):

In this study, the authors explored the antibacterial activity of two anthelmintics drugs (ivermectin and moxidectin) on a diverse set of 59 bacterial isolates. They compared their activity with 8 commonly used antibiotics and show that challenging some of these strains with the anthelmintic drugs can have an impact on antibiotic sensitivity. Overall the concept of the study is timely as the impact of non-antibiotics on diverse bacterial species starts to be unravelled and the potential cross-resistance between non-antibiotics and antibiotics is an emerging concept. However, the authors should re-consider some of the methodology they use to fit with what is generally done in the field. At the moment, very few conclusions can be taken from the data, which makes the study very descriptive. See below:

Major comments:

- These two drugs were part of the Maier et al., Nature, 2018 study, where they showed they had an activity on some gut microbes. In that study, the concentration used was 20 μ M and they only tested 40 gut species. Therefore the idea that these two drugs affect some bacterial species is not new, but here the authors used several drug concentration and another set of bacterial species.

We thank the reviewer for the positive overall appraisal of our study. We agree with the fact that ivermectin was part of the study published by Maier *et al.* in *Nature*, 2018. To the best of our knowledge, however, the presented study did not include moxidectin. Besides including moxidectin and the novelties mentioned by the reviewer, we want to emphasize that a key element of our study is the presented cross-reaction of ivermectin and moxidectin with antibiotics, especially considering large-scale administrations of these drugs. We clarified this in lines 89-97. Observed in the AUC

ratio measurements, repeated exposure to ivermectin or moxidectin *in vitro* changes the growth dynamics of a selection of isolates for a multitude of antibiotics. In this revision, we complemented the AUC ratio data with minimal inhibitory concentration (MIC) measurements for the anthelmintic challenged isolates of *S. salivarius*, *S. parasanguinis*, *S. mitis*, *S. dysgalactiae* and *S. pneumoniae* (see next comment).

- My major concern is the method to measure drug sensitivity/resistance based on AUC ratio. The whole antibiotics field typically uses the measure of MIC or IC90 to compare drug activities and call for sensitivity or resistance. Here, the authors only use the AUC ratio, which does not allow them to actually compare sensitivity or resistance between strains. Measuring MICs would allow to get quantitative data that can actually be compared.

We thank the reviewer for this important comment and addressed it with additional experiments. We determined MICs of 15 bacterial isolates in presence of erythromycin, clarithromycin, azithromycin, clindamycin, ciprofloxacin, tetracycline, meropenem and imipenem. The 15 tested bacterial isolates encompassed *S. salivarius* (01), *S. salivarius* (01-IVc), *S. salivarius* (01-MXc), *S. parasanguinis* (01), *S. parasanguinis* (01-IVc), *S. parasanguinis* (01-MXc), *S. pneumoniae* (02), *S. pneumoniae* (02-IVc), *S. pneumoniae* (02-MXc), *S. mitis* (01), *S. mitis* (01-IVc), *S. mitis* (01-MXc), *S. dysgalactiae* (01), *S. dysgalactiae* (01-IVc) and *S. dysgalactiae* (01-MXc).

While AUC ratios indicate a broad influence of anthelmintic treatment on antibiotic sensitivity, MICs only changed in specific cases. However, we would like to highlight three important points:

i) Bacterial exposure to ivermectin and moxidectin appears to alleviate delayed growth phenotypes, as evidenced by the increase in AUC ratio of the challenged isolate compared to the original. This suggests that the growth dynamics and antibiotic sensitivity can be distinct. Furthermore, changes in growth dynamics may already shift the compositional balance in the gut microbiome – given that this environment is highly competitive. We summarized this point in lines 304-321.

ii) Previous studies have shown an antibacterial effect of ivermectin on bacterial isolates. Building on this concept, our study reveals that anthelmintic challenging can rescue growth delays in the presence of these compounds. This finding strongly

indicates a bidirectional interaction between the bacterial isolate and the anthelmintic, which could impact treatment outcomes in helminth infections. Understanding these interactions is essential not only for addressing antimicrobial resistance but also for optimizing treatment efficacy against helminth infections, as discussed in lines 71-85 and the cited work by Schneeberger P.H.H., *et al.* (22).

iii) While the MICs for *S. salivarius*, *S. parasanguinis*, *S. pneumoniae*, and *S. dysgalactiae* remained largely unchanged, we observed significant alterations in MICs for *S. mitis* (01-IVc) with macrolide antibiotics and *S. mitis* (01-MXc) with AZ, as detailed in lines 249-256 and Table 3.

We added a section explaining the methodology of MIC testing: lines 481-501.

We added the following results concerning MICs: lines 249-256 and Table 3.

We expanded the discussion: lines 304-321.

- The challenging experiment is not clearly described in the text. How many passages have been done, for how many generation and at which drug concentration? How many parallel lineages are tested for each condition (and do they exhibit the same changes)? What is the bottleneck or dilution at each passage? Is resistance to anthelmintics actually reached (i.e. increased MIC after the challenge). My overall perception is that it is not clear whether what the authors describe comes from the cells affected from the first treatment and therefore becoming more "sensitive" or "resistant" to the second drug (drug combination effect) or if really the real resistance (increased MIC) to the first drugs make them more sensitive/resistant (collateral sensitivity or cross-resistance).

We regret the unclear description of the challenging experiment. To clarify, we i) added Supplementary Figure 1, which depicts the experimental procedure and ii) modified the description (lines 427-473), as well as Supplementary Table 4.

Minor comment:

- The rationale for choosing the 59 strains is not entirely clear to me.

We describe the rationale behind isolate selection in lines 353-389. The main focus lies on *Streptococcus* species, as a recent study by Schneeberger P.H.H., *et al.*, (22) found an association between treatment failure and presence of *S. salivarius*, amongst

others, in the participant's gut microbiome prior to albendazole-ivermectin combination treatment against STH infections (lines 362-364). Given their abundance throughout the GI-tract and the vastness of identified resistance mechanisms against macrolide antibiotics (lines 358-362), we wanted to investigate individual interactions in an expanded number of *Streptococcus* species.

Reviewer #3 (Remarks to the Author):

The study focuses on the antibiotic properties of two widely used anthelmintic medications and how they might affect the emergence of cross-resistance to macrolides and other antibiotic families, against various bacterial isolates.

Such a research topic is timely and relevant; however, I have a few comments to make:

Overall, it is correct that ivermectin and moxidectin are closely related broad-spectrum anthelmintics, both produced through fermentation by soil actinomycetes (*Streptomyces*). They also share some structural and physicochemical properties and a common mode of action. Therefore, considering the location of the target organism, we expect these two anthelmintic drugs to exhibit off-target effects and, to some extent, cross-react with other antibiotics. Additionally, perturbation of gut microbial growth is a complex process influenced by various factors including diet.

We thank the reviewer for this important comment. We acknowledge that, off-target effects to ivermectin and moxidectin are to be expected. However, we want to emphasize that studies i) exploring antibacterial properties of moxidectin, as well as ii) studies unravelling the difference in antibiotic susceptibility associated with repeated anthelmintic challenging, have not been published to date. We plan to verify our findings in a human cohort (stool sample metagenomics) and agree that other factors influencing gut microbial growth, such as diet, need to be factored in carefully. To clarify, we added the following lines (340-342) to the discussion:

“Furthermore, these molecular analyses should be extended to human cohort samples to detect whether off-target effects of the two anthelmintics actually manifest in a clinical setting.”

More specific comment:

1. Lines 55–57: Different settings have recorded varying responses to different anthelmintics. I suggest you include all human STHs with reported poor cure rates, as this study does not focus on a particular STH species.

We added the requested information in lines 54-57:

“Alarming, single-dose regimens of either drug yield poor cure rates in *Trichuris trichiura* infections and single-dose regimens of mebendazole treatment results in low cure rates against hookworm (*Ancylostoma duodenale*, *Necator americanus*) infections [7, 9, 10].”

Point-by-point response:

We thank the editor and the reviewers for providing the opportunity to revise and resubmit our manuscript. We are confident that the overall quality of our manuscript has further improved by addressing the suggestions put forth by all referees. Hence, we hope that it will now be suitable to be considered for publication in *Communications Biology*.

Our responses are highlighted in blue.

Reviewer #1 (Remarks to the Author):

The authors properly addressed my concerns, especially regarding the MIC data.

Reviewer #3 (Remarks to the Author):

Authors have addressed all comments raised in the previous version, except that supplementary Table 4 is missing from the revised paper.

We checked the Supplementary Data again and ensured that Supplementary Table 4 is both present and cited in the main manuscript file.